# Perceptions of Medical Students Regarding Career Counseling in Korea: A Qualitative Study

**DOI:** 10.3390/ijerph17103486

**Published:** 2020-05-16

**Authors:** Minsu Ock, Young-Joo Han, Eun Young Choi, Jeehee Pyo, Won Lee

**Affiliations:** 1Department of Preventive Medicine, Ulsan University Hospital, University of Ulsan College of Medicine, Ulsan 44033, Korea; 11351@naver.com (E.Y.C.); eesther0517@naver.com (J.P.); 2Department of Preventive Medicine, University of Ulsan College of Medicine, Seoul 05505, Korea; 3Korea Counseling Graduate University, Seoul 06722, Korea; 4Department of Nursing, Graduate School of Chung-Ang University, Seoul 06974, Korea; 5Department of Nursing, Chung-Ang University, Seoul 06974, Korea; oness38@daum.net

**Keywords:** medical students, career counseling, consensual qualitative research, focus groups, education, medical, graduate

## Abstract

Current medical school education focuses on acquiring appropriate knowledge with relatively little interest in developing the career selection skills of medical students. We investigated medical students’ perceptions of career problems and the required types of career counseling programs. Five focus group discussions were held with 23 medical students. The consensual qualitative study method was used to analyze the recorded discussion process. The medical students were more influenced by parents and grades than by subjective choices when deciding on admission to medical school. In future career choices, medical students considered the stability and feasibility of the career and expected quality of life. However, there were several opinions that it is essential to understand oneself. Objective and specific career information was lacking, and meeting with the professor was not very helpful for career counseling. Most medical students expected the effectiveness of the career counseling program but hoped the program would proceed with voluntary participation. Medical students wanted a variety of concrete and objective information, such as specialty information for choosing residency training, trainee hospital information, and post-residency training information in the career counseling program. Most medical students are not ready for career-related problems, therefore making it necessary to develop a career counseling program suitable for them.

## 1. Introduction

Contrary to the common belief that admission to a medical school determines one’s career, medical school students are required to select a specific area, such as general medicine or a clinical field, after completing 4–6 years of medical study [1]. Additionally, after becoming a medical resident, they must choose a specialty and training hospital. During this process, there is a frequent cessation of residency programs or change of specialty [2,3,4], which may represent the underdeveloped exploration of the desired career path during the specialty selection process. About 45% of physicians reported being satisfied with their life as a physician, while about 20% were unsatisfied [5]. Hence, it is necessary to examine if a deficiency in the careful consideration of career led to this circumstance.

In reality, students are overwhelmed with excessive academic competition and have no time to explore career options. Additionally, current medical school education focuses primarily on essential medical knowledge and skills, with relatively little interest in career choice and student counseling. Career decision-making for medical students should not be considered as a personal challenge for the students as their decisions affect their future patients [6]. Moreover, this decision will impact the life-long duty and performance of future physicians. Medical students with a high degree of adaptation to careers have high self-efficacy in their education [7]. Furthermore, 85% of students agreed regarding the need for career guidance in medical school [8]. Therefore, it is necessary to consider career counseling as one of the required courses for the medical school system [9]

Additionally, relevant research is deficient; only 25 articles with structured career programs and mentoring studies were published between 2000 and 2008 in PubMed [10]. Previous studies are looking at what specific majors medical students decide and what factors are relevant to the decision [11,12,13], but there is a lack of concern about how to help them choose their majors. It is also difficult to grasp in-depth preferences of medical students in career counseling, as most studies relied on the quantitative methodology of surveys [14]. In the Republic of Korea (hereinafter Korea), the development and operation of career counseling programs for medical students are scarce, and there is no evidence regarding its effectiveness.

Therefore, the purpose of this study is to establish a foundation for counseling interventions that assist medical students in choosing a suitable major or in aligning the direction of their career. Specifically, this study examines students’ recognition of their career and their need for a career counseling program.

## 2. Materials and Methods

In this study, focus group discussions were conducted with medical students to understand their career concerns. Furthermore, the focus group discussions focused on the necessity of career counseling programs as well as its components and operation procedure. In order to illustrate the critical themes of qualitative research, this research was described according to the Consolidated Criteria for Reporting Qualitative Research (COREQ) [15].

### 2.1. Criteria for Participant Selection, Recruitment, and Reflexivity Assessment

Participants in this qualitative study were fourth-year medical students and incoming fourth-year medical students, as the career issue would be most pertinent to them. Participants were recruited from A Medical University in Korea. The departmental student representative was informed of the purpose, method, and benefits of participating in the study, and the students who wish to participate were asked to contact the researchers directly. Students were explained in detail of their right to withdraw from the study, and their fully voluntary participation was emphasized.

### 2.2. Progression of the Focus Group Discussion

The focus group discussion was conducted in the classroom of the medical school according to semi-structured guidelines established by reviewing the previous research and discussions with researchers. The guidelines consisted of an ice-breaker, introduction/common questions, awareness of the need for a career counseling program, thoughts on the development of the program, and other discussions (Appendix A).

Five focus group discussions were conducted between November 2016 and February 2017. Two researchers participated as the facilitators of the discussion; a physician-researcher was the leading facilitator, and a nurse researcher was the assistant facilitator. The discussion lasted for about 70 to 90 min, and the entire discussion was recorded for the analysis.

### 2.3. Ethical Considerations

This study was approved by the Institutional Review Board of Asan Medical Center (IRB Number: 2016-1190). The participants provided written informed consent and were compensated with a 20,000 won (about 16 USD) coffee coupon and a lunch box worth 10,000 won (about 8 USD) each for participating in the study.

### 2.4. Analysis

The analysis was conducted using consensual qualitative research (CQR). CQR, which seeks to adopt the advantages of both quantitative and qualitative research methods, is widely used in the field of counseling [16,17]. In general, the CQR has an analysis team of three or more researchers that strives to increase the objectivity of the analysis and presents the core ideas of the participants as a quantitative cross-analysis table. In this study, CQR was used in consideration of strengthening objectivity in the analysis process and the field characteristics of the research subject. Transcriptions of the recorded discussion were used as data sources for analysis. The detailed data analysis was conducted according to the method proposed by Hill [16].

The analysis team consisted of five members. All five members had extensive research experience in qualitative research, and they all wrote their theses applying the qualitative research methodology. Two of them are master’s degree holders, and three are doctoral degree holders. One focused on researching the field of health care, two in nursing and health, and the other two in the field of counseling. One is a physician license holder, and two are a nurse license holder.

The analysis team held a total of six meetings from July 2017 to September 2017. They discussed their pre-conception and expectations on the research topic before the primary analysis. A professor with a major in counseling contributed as an auditor.

Based on the developed semi-structured guidelines, the analysis team established domains, sub-domains, and core ideas, followed by revising them through analyzing the first discussion. Each member of the analysis team then read the first and second discussions and coded them independently based on the developed domain and sub-domain. The results were discussed at the analysis meeting, and the coding results were modified. Afterward, one analysis team member, who had a doctorate in medicine, read the third, fourth, and fifth discussions and coded the data. The rest of the three members then reviewed and revised the coded data. Next, cross-analysis was performed to assign core ideas to domains. Core ideas that could not be categorized were placed in the ‘other’ category. The frequency information for each domain was calculated. In the case of frequency within the discussions, for each discussion, ‘+++’ were given if the domain was derived from three or more participants, ‘++’ if it was from two participants, ‘+’ if it was from one participant, and ‘-’ if it was from no one. For the overall frequency, the cases from all five groups were marked as general, more than three groups as typical, and less than three groups as a variant. The auditor confirmed the results of the cross-analysis, and the analysis team reviewed the feedback and revised it if necessary. However, the results were not confirmed by the participants.

## 3. Result

A total of 23 fourth-year medical school students participated in five focus group discussions (Table 1). A total of 313 semantic units were derived and categorized into 43 core ideas, seven sub-domains, and three domains. The cross-analysis results are shown in Table 2. The following sections describe the main contents of each domain.

### 3.1. The Motivation of Medical School Admission

#### Factors Affecting Medical School Admission

Among various factors influencing admission to medical school, the majority of participants referred to “the influence of parents” as the primary factor. Students with physician parents naturally made their admission decision as they grew up watching their parents’ technical aspects. On the other hand, students with non-physician parents determined to enroll in a medical school following parents’ wishes:
Participant 3-2: My mother strongly (suggested me to attend) a med school. (…) The goal of every mother in this country.

The opinions that medical school was chosen by the grades and the influence of surrounding people, such as teachers, were also mainly mentioned. For instance, some participants made the school choice as their grade was better than their expectancy; they did not want to discard the excellent grade, although they had another major in mind. In other words, the decision to enter the medical school was primarily determined by external factors rather than individual aptitude or interest.

### 3.2. The Reality of the Career Choice Process

#### 3.2.1. Factors That Affect Career Choice

Several elements affected the participants’ career choices after medical school. Of these elements, thoughts on the expected life quality, the career feasibility, and their understanding of themselves were demonstrated in all groups. Through the intensive academic process, they preferentially considered how much time they had to spend in their post-resident life, and likewise expected physical and mental fatigue. Therefore, predictable quality of life had a significant impact on their career choices. As they realized the unpleasant quality of life of the students in that specific department, they would now instead seek a specialty that guarantees more personal time. The participants who experienced physical and mental difficulties such as trauma after practicing in the preferred departments also changed their mind on their career paths after sensing the unsatisfactory expected quality of life, despite their interest in the field. Additionally, a female participant stated that a critical factor that persuaded her career choice was how the expected quality of life might affect marriage, childbirth, and parenting.


*Participant 3-3: For me, (it’s a) major that guarantees after-work life—securing personal time. You find happiness outside of your work. So, (a major) that promises a clock-out time.*



*Participant 4-6: Quality of life is important, and in the long run, as I am a woman, I will consider pregnancy, giving birth, and childcare as well.*


For participants, academic performance in medical schools was closely related to the feasibility of careers. It was considered that academic achievement has a direct relation with potential specialties, as it is often expressed that a specialty is a compromise of grades.


*Participant 2-4: You get to think like I could go there at least, or I could give a shot because you could see where you are at with your grade.*


However, several participants acknowledged the significance of self-understanding. They wanted to choose a specialty based on their talent or interest. Examples of these specialties based on talent and interest are consideration of the right skills for surgery, the capability to look into a microscope for a long time, the ability to interact with a patient well, compatibility with research, and many more. Also, when opening a medical practice after deciding on a specialty, the consideration of ability on management, accounting, and others was also accounted as critical factors for career decision-making:
Participant 2-1: It was aptitude; for excluding a ‘never-ever’ major, it was aptitude and interest (that played the criteria role).
Participant 2-2: It’s similar to me too. I exempted (specialties), and my aptitude, I mean I really can’t look into the microscope.

Typical core ideas included job satisfaction, career stability, and expected income. Moreover, a sense of rewarding at work—even with physical and mental fatigue, the visible performance of patients with treatment, and social recognition seem to influence job satisfaction:
Participant 5-3: I think it is essential to be recognized as much as you accomplish well. The recognition could be about the monetary term, but with the social climate, it’s important not to disregard any acknowledgment for the hard work.

#### 3.2.2. Concern over a Career Decision

Participants faced difficulties in choosing their careers because of objective and specific information deficiencies on career choice. Above all, many indicated that valuable information, including objective facts and prospects, was needed, rather than simple enumeration or hearsay information. One participant thought that the information given by a professor in a particular specialty might not be objective due to the professor’s affection for it. Some participants could not get any information about the specialties during the first- and second-year courses. Subsequently, they only had incomplete information during the third and fourth year of practice. In other words, they were being left in the dark:
Participant 1-4: I never got to know what was really going on in the hospital because during the first and second year, I was mostly in the classroom for lectures.

As a result, the majority of participants had vague ideas about careers without obtaining objective information. After all, many participants treated the choice of career as a simple matter. Some thought that notions on career could change over time. They believed that they could adjust themselves to the choice of their department; they did not consider their preference or their top-flight abilities, or they would simply switch to another specialty if one did not work. As one participant said, “Maybe it might be good to have an O.K. life, just as it flows,”—we can see that the issues of the medical school student’s career decision obscurely continues, hiding from the surface:
Participant 1-2: Some seniors said anything you do during med school is a waste because everything changes when you become an intern.

A short time to think of a career decision due to a busy academic life was also suggested as a typical core idea. Many participants explained that it was challenging to invest time to think about careers or to explore careers in the first and second years of medical school with an immense academic load. From the viewpoint of medical students who had piles of studying materials in front of their eyes, the concern and search for a career was not a priority:
Participant 3-3: It could sound like an excuse, but other than ‘Let’s finish up what’s under my nose’, you can’t think about anything because you have so much to study in med school. I heard a lot about (the career decision) when I first arrived at the school, but I need to focus on the pressing homework for now, and I guess I can go to another track if I have an opportunity later. I have a rough idea, but (regarding a detailed plan) for now, not really.

There were not enough people to share their concerns as well. Despite their parents having a significant influence on their admission to medical school, it was difficult for them to have meaningful discussions about their careers as their parents were not knowledgeable of the medical field in detail. Choosing a career became a competition among classmates at the end, which is why the participants had a hard time to voice their legitimate concerns with their classmates genuinely. In other words, everyone had to be wary around these classmates who claimed a spot in a specific department first. These practical challenges had naturally led to the desire of the students to have a career counseling program. However, several participants noted that no career counseling program was available despite the need for such:
Participant 4-4: I get worried that if a person says he or she would apply there, then should I just withdraw my application or should I go for it? So, I get agitated too.
Participant 4-1: Person who called dibs first seems to have the priority.

#### 3.2.3. Existing Career Counseling Practice

Meeting with an advisor was the prime channel related to career counseling. However, in-depth career counseling did not take place in the meeting, not even as the main topic, as the meeting was held only once or twice a year. The participants also indicated that many medical students were not able to partner up with the professors of the department they wanted. Mainly, as the advisor may be in a position to evaluate the students later, students were unable to talk freely and honestly. They also complained about difficulties in existing career counseling as they could not discuss their concerns about their career options. Moreover, students may face trouble if they decide to change their career path abruptly. We therefore confirmed that career counseling opportunities for the students were scarce:
Participant 2-1: Seeing my advisor is a bit… well, I was designated to him or her for that purpose, but if I were to be employed at that hospital, then our relationship may turn into an evaluating one, and I will have to tiptoe around if I later switch my interest from one to another…

There were other means of providing career information at school. There was also a curriculum that allowed students to learn about physicians not only in the medical field but also in different disciplines. However, this channel was insufficient for resolving the students’ questions about their careers, and many of the participants perceived the lecture as an open mic session for established physicians in each field to share their success stories:
Participant 3-3: I also thought like that because it was like an introduction to someone’s biography, rather than about the field. A majority of them spoke about their success stories, rather than actual information related to their field. It didn’t really help me. I just thought, ‘some people are just too cool’.

Student practice, classified as a typical core idea, also provided an opportunity for students to receive information about their careers. These enabled students to think about the career option available for them. Some claimed that what they read in books was different from the real world; as a consequence, they often change their preferred department during the practice. Career information acquisition such as information searching, meetings with seniors, and discussions with classmates were also categorized as typical core ideas. Participants said that they could obtain information on departments through an intern or resident, upper-level students from the same school, or from reading about it in an online community of physicians. However, even in this case, one participant reported that their doubts were not resolved as their classmates had the same amount of information as the participants:
Facilitator: Do you have the following type of conversations with your classmates?—“What department are you going to choose?”
Participant 5-5: Yes, I do.
Participant 5-2: Yeah, I do that a lot.
Facilitator: Then, does that resolve any of your doubts?
Participant 5-2: Well, we all have similar information.

### 3.3. Awareness of Career Counseling Program

#### 3.3.1. Expectations from the Career Counseling Program

Almost all participants expected the career counseling program to be effective. Through the career counseling program, they anticipated the following: a sense of accomplishment that they had learned their desired career, a more natural process for the department selection, and a decrease in career decision-making regrets. They also indicated that they were willing to participate in a prospective career counseling program. However, one participant raised concerns about the effectiveness of such a program. The existence of a career counseling program seemed to impose decision-making. The difficulty arises from the psychological pressure to discuss careers with classmates:
Participant 1-1: Most definitely helpful, but the regretful part is a separate issue—the further selection part. (It’s) For the peace of mind or that sort of thing.

#### 3.3.2. Contents of the Career Counseling Program

In the contents of the career counseling program, the necessity of providing department information and the direction of career after completing hospital training were classified as typical core ideas. Numerous participants wanted to know the actuality of each department rather than the superficial information they obtained from the practice. Many commented that they needed the information to determine their career direction after the completion of training. Specifically, there were requests of detailed numeric information such as the number of people remaining as professors after the residency, the number of people opening their practice, and the number of people entering other fields. There were also opinions that practical information on how difficult it was to open a private practice was needed. Visiting an alumnus or alumna who had opened a practice to inquire about the information was suggested as a way to achieve practical information:
Participant 1-2: I think it was constructive for the professor to come and show the career status of the alumni. Information regarding how many people are going to certain places, how many continue as professors, how many leave the hospital, and how many do other things will be helpful when you choose a department.

Some addressed that it was essential to provide information about prospective training hospitals, which were classified as a variant core idea. It was necessary to offer hospital information for students to acquire medical skills that will be useful in their medical practice. Information offering is essential as it ensures that those who do not gain hands-on experience from the hospital will still be sufficiently informed. The need for self-understanding and mentors was also categorized as a variant core idea. In particular, some participants with experience in psychological assessment anticipated the acknowledgment of self-interest as assisting the process of choosing their careers. However, the majority of participants who stated the requirement of self-interest obscurely considered the necessity of such activity. A mentoring session for career counseling during the practice was also suggested:
Participant 4-6: I went to a private counseling center to do the MBTI (Myers-Briggs Type Indicator); it was nice to get to know (more about) myself. I received some counseling—they offered it because I paid for the test and counseling.
Facilitator: How did it help you?
Participant 4-6: I am not sure about how it helped me along the way of choosing a department, but I vaguely—I don’t really remember it clearly—came to think about who I really was. The four alphabets (from MBTI) suited me; I think it’s nice to know yourself (better). I would have liked a psychological assessment (from the counseling).

#### 3.3.3. The Procedure of the Career Counseling Program

Various opinions were suggested regarding the procedure of the career counseling program; however, above all, many thought that the higher-grade students would be more equipped to participate in the career counseling program. Pre-medical course students or first- and second-year medical school students were perceived to be unsuitable for the program due to their heavy academic workload. Additionally, they were considered to have an insufficient amount of experience for career selection. Most of them insisted that the program was better suited to third- and fourth-year students:
Participant 3-1: I think the third and fourth years of med school would be good—the fourth year (preferably).

Many participants stressed the need for voluntary participation in career counseling programs. In other words, mandatory participation in the program would turn it into another part of the curriculum. This will result in counseling not being perceived as an opportunity to consider one’s career. One participant noted that if the program were operated with voluntary engagement, there would be no need to limit the timing or the target of the said program anymore:
Participant 1-3: If it’s based on voluntary participation, it would be really cool. If it has an enforcing engagement, then it becomes part of a curriculum, rather than being career exploration.

The wish to have fewer participants in the career counseling program was classified as a typical core idea. Several of them thought that a more customized program would be successful. Besides, they felt that it would be difficult to tell their stories honestly if many students were participating at the same time in the counseling session. Consequently, many claimed that it would be best to conduct a one-to-one counseling session instead:
Participant 1-3: I want a fewer number, but, realistically, eight should be the maximum.
Participant 1-4: I think it would be best if it were a one-to-one session, so, with maximum, three to four (people for the same program).

Participants also mentioned that instead of lecture-style career counseling, a round-table discussion would be better and more effective; this was classified as a typical core idea. Many participants believed that in the past, providing career information in a lecture format was not resourceful, and a round-table discussion will easily facilitate getting answers to questions. It was also assumed that such round-table discussions would enable personalized counseling. In terms of the frequency and duration of career counseling sessions, two to three sessions per program were considered appropriate; this was classified as a typical core idea. Additionally, a maximum of two hours per session was categorized as a variant core idea. There were thoughts that counseling over extended periods in long sessions would be burdensome for medical students who already lacked time to study:
Facilitator: If you do participate in a career counseling program, what would be an ideal number of sessions and an ideal length of time per session?
Participant 2-2: Two times.
Participant 2-1: One hour to one hour and a half.
Participant 2-2: Two times.
Participant 2-4: Twice, rather than once.
Participant 2-2: After the two sessions, a review (of the program) should be done.
Facilitator: Not really up for four times?
Participant 2-4: That’s all right.
Participant 2-1: I think that’s a bit much.

## 4. Discussion

The results of this study are significant as they provide a basis for counseling interventions that could help medical students choose a specialty that suits them. Among the findings of this study, it is worth noting that most of the factors that affect the admission of medical students were external. Despite the consideration of a career’s attractiveness and the student’s academic disposition, students chose to study medicine due to external factors, such as their parents’ influence, grades, and job security, rather than personal inclinations. Besides, parental influences were derived as a general core idea. This finding is in contrast to previous studies, which indicated that interest and aptitude were considered as the priority for medical school entrance [18]. The gradually increasing minimum entrance grade can explain this contradiction for medical schools and the parents’ interest in the medical school admission as compared to what we had fifteen years ago.

As Kim and her colleagues claimed, the influence of parents on medical students’ lives is considered to be significant [19]. Accordingly, students may have low self-efficacy in career decision-making. As self-efficacy is a crucial element of career choice and growth [20,21], it is necessary to evaluate career decision self-efficacy in the career counseling program for medical students. The factors that impact career choice after the entrance seem to be little influenced by parents. Therefore, medical students are expected to have a greater need for mentors or third-party advisors other than their parents to provide career counseling [22].

Previous research in Korea identified family, friends, professors, and experiences of patient interaction and school practice as prime career determinants [23]. According to a systematic review that synthesized studies on factors related to the selection of majors by medical students, academic interests, competencies, controllable lifestyles, or flexible work schedules were identified as key factors [14]. However, in this study, the central factors prompting career choices were presented: career feasibility, the expected quality of life, and self-understanding. Besides, it is noticeable that self-understanding was derived, which was not ascertained among the components of motivation for the medical school entrance. It appears to be positively construed that the self-understanding was classified as a leading element for influencing the career decision-making, as it was the most basic procedure for selecting a career [24,25]. Meanwhile, the factors of career feasibility, such as grades, were still categorized as significant factors. This factor was not reported in other countries’ prior results. It can be expected that there may be cultural and social differences in medical students’ career choices. Furthermore, the expected quality of life has been suggested as a critical factor for medical students. They are likely to experience high academic loads and chronic academic burnouts [26,27]. This would cultivate a desire for a personal time after witnessing the intensely occupied lives of residents.

It is also noteworthy that the lack of objective and specific information regarding the careers was presented as a significant concern for career choice. Therefore, for effective career counseling, it is essential to provide a variety of information such as career-related materials, information on career direction after completion of training, and hospital information [28]. Nevertheless, the range of career-related information is so extensive that it is indispensable for preparing a system for medical students to have easy access to the information. The qualities and abilities required in these specific areas, and the salaries and values that can be obtained, vary considerably [8]. Most of all, career information may need to be updated regularly as its use may change over time. Therefore, it is a useful measure to arrange career-related information on the school website and let the students collect additional information and share these among their classmates, thus creating a wide pool of information that is readily available [29].

The most frequently mentioned existing career counseling was meeting with an advisor. In-depth career counseling, unfortunately, was not indicated in this research, as the results of previous studies demonstrated that meetings with advisors were considered mere friendship [9]. Thus, medical schools should not acknowledge meeting with the advisor as ‘career counseling’ and student life guidance. The schools should implement a supervisor system, wherein a supervisor will provide advice to subordinates such as students for practical career counseling. Also, it is suggested to pursue other resources, such as upperclassmen and classmates, who can provide career counseling and a more thorough discussion about careers.

It will be essential to reflect on what the career counseling program will contain and how to go about it with such a program. Moreover, as this study confirmed that it is crucial to operating a career counseling program with voluntary participation, it must enhance medical students’ self-efficacy regarding career exploration and decision-making. Besides, medical students should be able to assess their occupational values and use them in the career selection process [9,27,30]. At this time, it should be considered that the academic stress of medical students may also affect the operation of career counseling programs [26,31]. Furthermore, to generate a more significant effect on career counseling, a career counseling program should be based on counseling theory [32].

When developing a career counseling program for medical students, it is anticipated to contemplate group counseling methods in consideration of various advantages of group counseling, such as effectiveness and efficiency [32,33]. However, it is necessary to take into account the fact that several participants in this study mentioned that only a few people participated in career counseling programs, and some even preferred one-to-one counseling. This result conforms with those of previous studies suggesting that one-to-one counseling is preferred among medical students for mentoring, rather than group meetings such as round-table discussions [34]. Moreover, it is essential to consider that there is a competition between classmates in medical schools. As a result, they cannot be honest about their career concerns in group counseling. Therefore, it is important to consider forming a group of people who do not overlap with their preferences of career for the program or encouraging students to form their groups for peer counseling. Moreover, as the students have limited time, it is estimated that securing sufficient career counseling time for individuals will increase the satisfaction of career counseling programs.

Lastly, career counseling programs for medical students should be prioritized for upperclassmen, such as third- and fourth-year students. In other words, the prime target of the career counseling program should be those for whom career decision-making is a more practical and pressing issue. Such a counseling program would be unsuitable for first- and second-year medical school students who are overwhelmed with the academic load. Previous studies also recommended that it is crucial to prepare a student guidance program that focuses on school life and academic counseling for pre-medicine students. It is also suggested to prepare academic counseling and emotional counseling for the first- and second-year students and career counseling for the third- and fourth-year students [9].

This study has the following limitations. First, the recruitment of the study was performed at only one medical school. Thus, the results of the study may reflect the characteristics of one medical school only. Second, because one researcher who was mainly involved in participant recruitment was an alumnus of the medical school, the possibility that research participants felt compelled to participate in the study cannot be entirely excluded. Efforts were made to ensure that participation of the students was voluntary; however, in terms of reflectivity, the possibility that the participants spoke in the taste of the researchers could not be entirely excepted. Third, methodological limitation of focus group discussions may be pointed out. Focus group discussions have the advantage of being able to collect balanced opinions through interactions among members, but they are likely not to reveal an individual’s candid mind. In a study focusing on competition in career issues among medical students, it would be better to use in-depth interview techniques than focus group discussions.

## 5. Conclusions

Based on the results, a career counseling program for medical students is required for promoting their self-examination of their preferences and understanding of their abilities and providing students with information about department and career direction after training. Additionally, courses for students to explore hospital- and career-related information are required. Further research is needed to develop and apply a career counseling program for medical students, and to evaluate its effectiveness. Through this process, the medical students’ challenges regarding career selection can be identified, and the program can be accordingly modified and supplemented.

## Figures and Tables

**Table 1 ijerph-17-03486-t001:** Characteristics of the research participants.

No.	Participant Number	Gender	Focus Group Discussion Number
1	1-1	M	Group 1
2	1-2	M	Group 1
3	1-3	M	Group 1
4	1-4	F	Group 1
5	2-1	M	Group 2
6	2-2	M	Group 2
7	2-3	M	Group 2
8	2-4	M	Group 2
9	3-1	M	Group 3
10	3-2	M	Group 3
11	3-3	F	Group 3
12	3-4	M	Group 3
13	4-1	M	Group 4
14	4-2	M	Group 4
15	4-3	M	Group 4
16	4-4	F	Group 4
17	4-5	M	Group 4
18	4-6	F	Group 4
19	5-1	M	Group 5
20	5-2	M	Group 5
21	5-3	M	Group 5
22	5-4	M	Group 5
23	5-5	M	Group 5

**Table 2 ijerph-17-03486-t002:** Results of the cross-analysis.

Domain	Sub-Domain	Core Idea	Frequency ^1^					Total Frequency ^2^
			Group 1	Group 2	Group 3	Group 4	Group 5	
1. The motivation for medical school admission	1.1. Factors affecting medical school admission	1.1.1. Influence of teacher and surrounding people	+	-	+++	+	-	Typical
		1.1.2. Influence of parents	++	+++	++	+++	+++	General
		1.1.3. Selected according to grades	++	+	+++	-	+++	Typical
		1.1.4. Selected for job stability	+	-	+	++	+	Typical
2. Career Choice Process	2.1. Factors affecting career choice	2.1.1. Expected income	-	+	+	-	+++	Typical
		2.1.2. Expected quality of life	++	+	+++	++	+++	General
		2.1.3. Job satisfaction	++	-	+	-	+++	Typical
		2.1.4. Career feasibility	++	+	++	+	+++	General
		2.1.5. Self-understanding	+++	+++	+++	+	+	General
		2.1.6. The stability of job	+++	+++	+	-	-	Typical
	2.2. Concern over a career decision	2.2.1. Obscure thoughts on career	+++	++	+	+++	+	General
		2.2.2. Absence of career counseling program	+	+	+++	-	-	Typical
		2.2.3. Unable to think about career due to heavy academic workload	+++	-	+	+	-	Typical
		2.2.4. Deficiency in objective and detailed information	+++	+++	++	++	+	General
		2.2.5. Deficiency in people to share concerns	+++	+	-	+	-	Typical
		2.2.6. Absence of career concern	++	-	+	+	-	Typical
		2.2.7. Competition in the career choice process	-	-	+	++	+	Typical
	2.3. Existing career counseling practice	2.3.1. Meeting with an advisor	+++	+++	+++	+++	+++	General
		2.3.2. Information research for career information acquisition	+++	++	-	-	+	Typical
		2.3.3. Career information provided by school	++	-	+++	+++	+++	Typical
		2.3.4. Meeting with upperclassmen	-	-	+	+++	+	Typical
		2.3.5. Career exploration through student practice	-	+	+	+	+++	Typical
		2.3.6. Career discussion with classmates	-	-	++	+++	+++	Typical
3. Awareness of career counseling program	3.1. Expectations from the career counseling program	3.1.1. Expectancy of program effectiveness	+++	+++	+++	++	++	Typical
		3.1.2. Apprehension of program effectiveness	-	+	-	-	-	Variant
	3.2. Contents of the career counseling program	3.2.1. Necessity of department information provision	++	++	+	+	-	Typical
		3.2.2. Necessity of hospital information provision	+++	+	-	-	-	Variant
		3.2.3. Necessity of career direction after training information provision	+++	+	-	+++	++	Typical
		3.2.4. Necessity of a mentor	+	+	-	-	-	Variant
		3.2.5. Necessity of self-understanding	+++	-	-	+++	-	Variant
	3.3. The procedure of the career counseling program	3.3.1. Suitable eligibility of upperclassmen	+++	+++	+++	++	+++	General
		3.3.2. Necessity for voluntary involvement	+++	++	+	-	+	Typical
		3.3.3. Adequacy of the number of sessions for the program—two to three	++	+++	-	-	+	Typical
		3.3.4. Suitable with fewer number of participants	+++	+++	+	++	-	Typical
		3.3.5. Suitable with the round-table format than a lecture	++	+	+	-	-	Typical
		3.3.6. Adequacy of the length of each—maximum of two hours and half	+++	+	-	-	-	Typical

^1^-: Zero participant, +: One participant, ++: Two participants, +++: More than three participants. ^2^ General: All group (five groups), Typical: More than half (three groups) of the total groups, Variant: Less than half (three groups) of the total groups.

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
