# Peer review of "Perceptions of Medical Students Regarding Career Counseling in Korea: A Qualitative Study"

_ijerph, 2020, doi:10.3390/ijerph17103486_

Round 1
Reviewer 1 Report
Thank you for asking me to review the manuscript for International Journal of Environmental Research and Public Health, “Perceptions of medical students regarding career counseling in Korea: a qualitative study.” The paper is a needs analysis and a report of a qualitative research project into the perceptions of career counselling by Korean medical students. I think the authors may have come up with some interesting findings but a number of issues need to be addressed before publication.
Why did they use CQR? How useful was it as a method? Why did they choose this over other theoretical approaches or methods?
It is clear that English is a second language for the authors. There are occasional grammatical errors but also many statements that are grammatically correct but awkward and difficult to understand. For example, the statement “Moreover, a sense of rewarding at work …” does not make sense until one realizes that the authors might be trying to say that the participants discussed work that was personally rewarding?
Another example is from the translation of the quotes. “I think it is essential to be recognized as much as you accomplish well” makes no sense.
Another example in the Discussion is the following sentence. It is grammatically correct but meaningless. “Besides, it is noticeable that self-understanding was derived, which was not ascertained among the components of motivation for the medical school entrance”.
The authors take a lot of trouble to establish the credentials of the researchers which is not usually necessary.
Other medical schools in the world do have career counselling services for students and have had them for some time. It is not clear what knowledge the authors have of existing services.
They need to make it clear that this is very much a study localized to the Korean context. There are some cultural issues that might need to be discussed. It seems to be an underlying assumption that people who go to medical school in Korea do so because their parents decide that they will. Many readers from other cultures, where parental preferences are less of an influence, would find this strange. It could be that subsequent dissatisfaction with a medical career is due to the fact that the student did not choose to go to medical school but went because their parents made the decision for them. Likewise, the statement that “Students with physician parents naturally made their admission decision as they grew up watching their parents’ technical aspects.” sounds very strange to a Westerner. Most Westerners would assume that if the children of physicians chose to pursue a medical career it was because they wanted the lifestyle and the decision had little to do with “technical aspects”.
One student mentions getting a Myers-Brigg test. Do the authors critique this? There is a view that tests, like this, have no validity.
The authors need to be careful in how they run focus groups and discuss their findings. Normally, these should be run to hear participants share their experience. In this study, though, the authors also asked these participants to speculate about how a potential career counselling service should work. While this might be useful and interesting information it does not carry the same weight as stories the participants might tell about their concrete experience. The discussion should be more critical when considering such speculations.
It is not clear why the information in the tables has been provided. It is essentially irrelevant in a qualitative study.
Author Response
Reviewer 1
Thank you for asking me to review the manuscript for International Journal of Environmental Research and Public Health, “Perceptions of medical students regarding career counseling in Korea: a qualitative study.” The paper is a needs analysis and a report of a qualitative research project into the perceptions of career counselling by Korean medical students. I think the authors may have come up with some interesting findings but a number of issues need to be addressed before publication.
Response: We would like to thank you for reading our manuscript and reviewing it. The followings are point-by-point responses to your comments.
Why did they use CQR? How useful was it as a method? Why did they choose this over other theoretical approaches or methods?
Response: In this study, CQR was used among various qualitative analysis methodologies. CQR, which seeks to adopt the advantages of both quantitative and qualitative research methods, is widely used in the field of counseling. In addition, the CQR has an analysis team of three or more researchers that strives to increase the objectivity of the analysis and presents the core ideas of the participants as a quantitative cross-analysis table. In this study, CQR was used in consideration of strengthening objectivity in the analysis process and the field characteristics of the research subject. The reasons for adopting CQR as an analytical method in this study are added to the Method section (Line 95~).
It is clear that English is a second language for the authors. There are occasional grammatical errors but also many statements that are grammatically correct but awkward and difficult to understand. For example, the statement “Moreover, a sense of rewarding at work …” does not make sense until one realizes that the authors might be trying to say that the participants discussed work that was personally rewarding? Another example is from the translation of the quotes. “I think it is essential to be recognized as much as you accomplish well” makes no sense. Another example in the Discussion is the following sentence. It is grammatically correct but meaningless. “Besides, it is noticeable that self-understanding was derived, which was not ascertained among the components of motivation for the medical school entrance”. The authors take a lot of trouble to establish the credentials of the researchers which is not usually necessary.
Response: Although we have already received English editing, we have received additional English editing. We also reread the entire manuscript and made it easy to understand. Nevertheless, we think the sentences you pointed out can be interpreted as cultural differences. There seems to be a difference in the perception of medical students from Western cultures and those from Eastern cultures like Korea. We think this study has highlighted that difference.
Other medical schools in the world do have career counselling services for students and have had them for some time. It is not clear what knowledge the authors have of existing services.
Response: As you suggested, in the introduction, related prior studies, including foreign literature, were further reviewed and added to highlight the limitations of previous studies and the need for this study (Line 55~).
They need to make it clear that this is very much a study localized to the Korean context. There are some cultural issues that might need to be discussed. It seems to be an underlying assumption that people who go to medical school in Korea do so because their parents decide that they will. Many readers from other cultures, where parental preferences are less of an influence, would find this strange. It could be that subsequent dissatisfaction with a medical career is due to the fact that the student did not choose to go to medical school but went because their parents made the decision for them. Likewise, the statement that “Students with physician parents naturally made their admission decision as they grew up watching their parents’ technical aspects.” sounds very strange to a Westerner. Most Westerners would assume that if the children of physicians chose to pursue a medical career it was because they wanted the lifestyle and the decision had little to do with “technical aspects”.
Response: As you pointed out, this study seems to have highlighted the cultural and social differences in medical students’ career choices. We reviewed more relevant previous studies and added them to the Discussion section.
One student mentions getting a Myers-Brigg test. Do the authors critique this? There is a view that tests, like this, have no validity.
Response: Thank you for your comments. However, qualitative research is aimed at reflecting participants’ opinions as they are, so the study did not criticize or offer informative feedback on the participants’ remarks. In this study, the main opinions of the participants were expressed as they were.
The authors need to be careful in how they run focus groups and discuss their findings. Normally, these should be run to hear participants share their experience. In this study, though, the authors also asked these participants to speculate about how a potential career counselling service should work. While this might be useful and interesting information it does not carry the same weight as stories the participants might tell about their concrete experience. The discussion should be more critical when considering such speculations.
Response: We agree entirely with your point. So in this study, we focused on listening to the participants’ experiences and opinions on career counseling at the beginning of the focus group discussion, and only in the latter part did we confirm their opinions on the necessity and components of the career counseling program. We already described these points in the Method section (Line 82). Furthermore, we conducted focus group discussions, careful not to influence the opinions of the participants with our intentions.
It is not clear why the information in the tables has been provided. It is essentially irrelevant in a qualitative study.
Response: In CQR, the cross-analysis table is presented as the primary study result. Please refer to the references related to CQR.

Reviewer 2 Report
The Authors have presented an interesting paper. This manuscript addresses an important issue of perception of medical students regarding career counseling. The topic and objectives of the study is significant for the readers of International Journal of Environmental Research and Public Health. However, the paper requires few corrections and additional explanations.
Introduction:
- Line 38-39: “A total of 40% 38 of physicians reported being satisfied with their life as a physician, while 20% were unsatisfied” - the authors cited source from 2002 (nearly 20 years ago). Authors should cite more recent references.
- Line 51-52: “Additionally, there is a deficiency in relevant research; only 25 articles with structured career 51 programs and mentoring studies were published between 2000 and 2008 in PubMed”. How about 2008-2020? Authors should expand review of current literature.
Materials and Methods:
- When the focus group discussion was conducted? Please define timeframe.
- The semi-structured guidelines (focus group scenario) should be described in details and presented as a supplementary file.
- Five focus group discussions were conducted and total 23 students were participated. The number of participants in focus group was 4-5 students, so all of them were mini focus group. Please refer to this fact and comment it in methods section and limitations of the study. You can based on the article of Nyumba et al. (2018) titled “The use of focus group discussion methodology: Insights from two decades of application in conservation”.
Available from:
https://besjournals.onlinelibrary.wiley.com/doi/epdf/10.1111/2041-210X.12860
Discussion:
- In the Discussion part - for international readers – should be supported by more findings from international research.
Author Response
Reviewers comment
Reviewer 2
The Authors have presented an interesting paper. This manuscript addresses an important issue of perception of medical students regarding career counseling. The topic and objectives of the study is significant for the readers of International Journal of Environmental Research and Public Health. However, the paper requires few corrections and additional explanations.
Response: We would like to thank you for reading our manuscript and reviewing it. The followings are point-by-point responses to your comments.
Introduction:
Line 38-39: “A total of 40% 38 of physicians reported being satisfied with their life as a physician, while 20% were unsatisfied” - the authors cited source from 2002 (nearly 20 years ago). Authors should cite more recent references.
Response: As you suggested, we revised it with the latest article.
Line 51-52: “Additionally, there is a deficiency in relevant research; only 25 articles with structured career 51 programs and mentoring studies were published between 2000 and 2008 in PubMed”. How about 2008-2020? Authors should expand review of current literature.
Response: Unfortunately, we were not able to find any newer articles that are similar to that article. Instead, in the introduction, related prior studies, including foreign literature, were further reviewed and added to highlight the limitations of previous studies and the need for this study (Line 55~).
Materials and Methods:
When the focus group discussion was conducted? Please define timeframe.
Response: The focus group discussion was held five times between November 2016 and February 2017 (Line 85).
The semi-structured guidelines (focus group scenario) should be described in details and presented as a supplementary file.
Response: As you suggested, we attached a semi-structured guideline as a supplemental document.
Five focus group discussions were conducted and total 23 students were participated. The number of participants in focus group was 4-5 students, so all of them were mini focus group. Please refer to this fact and comment it in methods section and limitations of the study. You can based on the article of Nyumba et al. (2018) titled “The use of focus group discussion methodology: Insights from two decades of application in conservation”.
Available from: https://besjournals.onlinelibrary.wiley.com/doi/epdf/10.1111/2041-210X.12860
Response: As you pointed out, the number of participants per group in these focus group discussions is 4 to 5. According to the suggested article, a mini focus group needs to be composed of individuals with a high level of expertise. However, the purpose of this study is to examine the perception of career counseling for current and incoming fourth-year medical students, so it is considered that there is no particular expertise required. Therefore, whether it is a mini focus group does not appear to be a limitation of this study.
Discussion:
In the Discussion part - for international readers – should be supported by more findings from international research.
Response: As you pointed out, we added foreign references to support our findings and claims.

Round 2
Reviewer 1 Report
Thank you for asking me to review the resubmitted manuscript for International Journal of Environmental Research and Public Health, “Perceptions of medical students regarding career counseling in Korea: a qualitative study.” I think the authors have some valuable insights but there are still issues with the manuscript.
The standard of English is generally very good. However, it is clear that English is a second language and many sentences still come across as awkwardly expressed, even when grammatically correct. There are still some grammar errors.
The emphasis needs to be more on the inner experiences of the individuals involved, such as their doubts and personal struggles in making career decisions. The pioneers of CQR claim that it is a method that is ideal for studying the inner experiences of people. It seems that there are advisors who do meet with the students but there are a number of problems with the current system. The participants expressed some of their difficulties in relating to the advisors. I think the readers might like to hear more about these difficulties as well as more about the organization of the current system, as other countries have similar advisory arrangements. The need for better career counseling may be a conclusion of the study but it is not the focus of the study. The focus needs to be the internal struggles of the students. At present, the possible solution of a career counseling service seems to take up too much of the paper, especially in the discussion.
For the future, the authors might choose to do one-on-one interviews as the focus groups seemed to raise sensitive issues. For example, the authors found out that “Choosing a career became a competition among classmates … which is why the participants had a hard time to voice their legitimate concerns with their classmates genuinely”. It might be that a focus group setting could inhibit some students from articulating these concerns and this needs to be admitted as a limitation of the study.
The authors state that they selected CQR because it is a qualitative approach that allows for objectivity. For me, this is troubling. Qualitative research tries not to achieve objectivity and/or reliability but understanding of the subjective perceptions of the participants. Despite the claims for CQR, I do not believe such objectivity is attainable or even desirable. However, I admit that this comes down to philosophical differences and understandings of what different kinds of research can achieve. For the future, if the authors wish to engage in more qualitative research then I suggest they engage with more mainstream theoretical backgrounds. I would recommend texts such as Cleland & Durning (2015) Researching Medical Education. Wiley Blackwell. If they wish to engage in mixed methods research that really does combine the best of qualitative and quantitative approaches then I would recommend the work of scholars such as John and J David Cresswell (2017) Research Design: qualitative, quantitative, and mixed methods approaches. 5th ed. SAGE.
Author Response
Reviewer 1
Thank you for asking me to review the resubmitted manuscript for International Journal of Environmental Research and Public Health, “Perceptions of medical students regarding career counseling in Korea: a qualitative study.” I think the authors have some valuable insights but there are still issues with the manuscript.
Response: We would like to thank you for reading our manuscript and reviewing it. The followings are point-by-point responses to your comments.
The standard of English is generally very good. However, it is clear that English is a second language and many sentences still come across as awkwardly expressed, even when grammatically correct. There are still some grammar errors.
Response: We already received English editing twice, but we read the entire manuscript again and checked for grammar errors.
The emphasis needs to be more on the inner experiences of the individuals involved, such as their doubts and personal struggles in making career decisions. The pioneers of CQR claim that it is a method that is ideal for studying the inner experiences of people. It seems that there are advisors who do meet with the students but there are a number of problems with the current system. The participants expressed some of their difficulties in relating to the advisors. I think the readers might like to hear more about these difficulties as well as more about the organization of the current system, as other countries have similar advisory arrangements. The need for better career counseling may be a conclusion of the study but it is not the focus of the study. The focus needs to be the internal struggles of the students. At present, the possible solution of a career counseling service seems to take up too much of the paper, especially in the discussion.
Response: As we already described in Introduction section, the purpose of this study is to establish a foundation for counseling interventions that assist medical students in choosing a suitable major or in aligning the direction of their career. Specifically, this study focused medical students’ recognition of their career and their need for a career counseling program. So, of course, these are the main contents of this study. As you have pointed out, it would also be meaningful to conduct qualitative research focusing on the internal stressgles of medical students. It would be a good idea to address this in further research.
For the future, the authors might choose to do one-on-one interviews as the focus groups seemed to raise sensitive issues. For example, the authors found out that “Choosing a career became a competition among classmates … which is why the participants had a hard time to voice their legitimate concerns with their classmates genuinely”. It might be that a focus group setting could inhibit some students from articulating these concerns and this needs to be admitted as a limitation of the study.
Response: Thank you for your careful comment. As you suggested, we incuded methodological limitation of focus group discussions.
The authors state that they selected CQR because it is a qualitative approach that allows for objectivity. For me, this is troubling. Qualitative research tries not to achieve objectivity and/or reliability but understanding of the subjective perceptions of the participants. Despite the claims for CQR, I do not believe such objectivity is attainable or even desirable. However, I admit that this comes down to philosophical differences and understandings of what different kinds of research can achieve. For the future, if the authors wish to engage in more qualitative research then I suggest they engage with more mainstream theoretical backgrounds. I would recommend texts such as Cleland & Durning (2015) Researching Medical Education. Wiley Blackwell. If they wish to engage in mixed methods research that really does combine the best of qualitative and quantitative approaches then I would recommend the work of scholars such as John and J David Cresswell (2017) Research Design: qualitative, quantitative, and mixed methods approaches. 5th ed. SAGE.
Response: Thank you for suggesting additional reading materials. The CQR's pursuit of objectivity and/or reliability is more of a methodological rigor than a study's outcome. We think the charm of qualitative research is that it is possible to combine various methodologies and perspectives. Based on your opinion, we will continue to conduct qualitative research based on various theoretical backgrounds.

Reviewer 2 Report
Thank you. The manuscript has been significantly improved.
Author Response
Reviewer 2
Thank you. The manuscript has been significantly improved.
Response: We would like to thank you for reading our manuscript and reviewing it.
